# Postoperative Complications of Hip Fractures Patients on Chronic Coumadin: A Comparison Based on Operative International Normalized Ratio

**DOI:** 10.3390/geriatrics5030043

**Published:** 2020-07-15

**Authors:** Michael S. Kain, David Saper, Kyle Lybrand, Kasey-Jean Bramlett, Paul Tornetta III, Peter Althausen, John S. Garfi, Donald P. Willier III, Ruijia Niu, Andrew J. Marcantonio

**Affiliations:** 1Department of Orthopaedic Surgery, Boston Medical Center, 850 Harrison Avenue, Boston, MA 02118, USA; paul.tornetta@bmc.org (P.T.III); ruijia.niu@bmc.org (R.N.); 2Orthopaedic and Rehabilitation Centers, 5616 North Western Avenue, Chicago, IL 60659, USA; dave.saper@gmail.com; 3Ortho Montana, 2900 12th Avenue North, Billings, MT 59101, USA; kylybrand@gmail.com; 4Department of Orthopaedic Surgery, Lahey Hospital & Medical Center, 41 Mall Road, Burlington, MA 01805, USA; kasey.bramlett@lahey.org (K.-J.B.); John.Garfi@Lahey.org (J.S.G.); dpw012@jefferson.edu (D.P.W.III); Andrew.J.Marcantonio@Lahey.org (A.J.M.); 5Reno Orthopaedic Clinic, Reno, NV 89503, USA; peteralthausen@outlook.com

**Keywords:** hip fracture, INR, anticoagulation, postoperative complications

## Abstract

In current clinical practice, orthopedic surgeons often delay the surgery intervention on geriatric hip fracture patients to optimize the international normalized ratio (INR), in order to decrease the risk of postoperative hematological complications. However, some evidence suggests that full reversal protocols may not be necessary, especially for patients with prior thromboembolic history. Our study aims to compare the surgical outcomes of patients with normal versus elevated INR values. We conducted a retrospective chart review on 217 patients who underwent surgeries on hip fractures at two academic trauma centers. We found that in our group (n = 124) of patients with an INR value of 1.5–3.0, there was only one reoperation for a hematoma, but there was a trend for more blood transfusions. There was no statistically significant difference in the odds of reoperation or overall complications. Nevertheless, there were significantly more events of postoperative anemia in this high INR patient group.

## 1. Introduction

Early operative intervention of hip fractures has demonstrated a decrease in both morbidity and mortality in the geriatric population [1,2,3,4,5,6,7,8]. However, orthopedic surgeons often delay surgical interventions to optimize therapeutic international normalized ratio (INR) value of patients, a protocol aimed at decreasing the risk of bleeding complications and improving overall outcomes [9,10,11]. The clinical practice of delay is based on the philosophy that decreasing the INR at the time of surgery will decrease estimated blood loss (EBL), transfusion risk, and bleeding complications [12]. There is some evidence to contradict this, as some authors have suggested operating through antiplatelet therapy [13] or only give a single dose of vitamin K [14,15]. Yet, the risk of peri-operative bleeding and complications when operating on patients with elevated INR for hip fractures is relatively unknown.

In the general population of the US, there is a 1.5% prevalence of chronic therapeutic anticoagulation and this is even higher in the geriatric population [16]. In the hip fracture population, it has been estimated that a minimum, 4–8% of hip fractures are on chronic coumadin [17]. Reversing or discontinuing therapeutic anticoagulation is not without potential complications. Patients with atrial fibrillation or a history of prior thromboembolic event have a lower risk of an adverse event from a temporary discontinuation of warfarin compared to patients who have a mechanical heart valve [18,19,20,21,22,23]. The thought behind avoiding anticoagulation reversal to a sub-therapeutic range (an INR value below 1.5) is to avoid a supra-therapeutic ‘rebound’ INR from warfarin therapy, which theoretically leads to a bleeding complication. Avoiding reversal also prevents the need for large therapeutic bridging doses of low molecular weight heparin products that could also lead to increased bleeding, and a prolonged period of a sub-therapeutic INR values post-operatively and could elevate the risk of a thromboembolic event. Additionally, the morbidity associated with surgical delays may be avoided [1,2,5,6,7,8,12,24].

At our institutions, patients who sustained hip fractures were routinely taken to the operating room for open reduction internal fixation, cephalomedullary nailing, or arthroplasty procedures without an aggressive reversal protocol and at times with no reversal if the INR was less than three. This practice has evolved over time and thus the purpose of this study was to evaluate the perioperative outcomes and safety of surgical intervention in hip fracture patients with therapeutic and sub-therapeutic INR values. Our hypothesis was that there would be no difference in complication rate following surgical intervention for patients with therapeutic versus sub-therapeutic INR values.

## 2. Materials and Methods

### 2.1. Study Patients

We performed an IRB-approved retrospective chart review of patients who underwent surgical intervention of hip fractures from 2005 to 2013 at two level II trauma centers. All geriatric hip fracture patients aged 65 years or older and who were on chronic warfarin were included in our study. All hip fracture types (femoral neck, intertrochanteric, and subtrochanteric fractures) were included. Surgical interventions including closed reduction percutaneous screw fixation (CRPP), open reduction internal fixation (ORIF), short and long cephalomedullary nailing (CMN), and arthroplasty were performed. Patients were excluded if their fracture was treated with non-operative management, or had an INR greater than 3.0 at the time of surgical intervention.

### 2.2. Data Collection

Baseline data including age, type of fixation, days to surgery, INR at the time of surgery, and the use of vitamin K and/or fresh frozen plasma (FFP) were reviewed. We used the INR value measured immediately before surgery as the INR value at the time of surgery. Patients were grouped into low and high INR groups if they had a therapeutic INR value below 1.5 or between 1.5 and 3.0, respectively.

The primary outcomes were the need for re-operation within 30 days for drainage of a postoperative hematoma or a major medical complication including cardiac events (atrial fibrillation, myocardial infarction, congestive heart failure), infection (urinary tract infection, pneumonia), thromboembolic complication, or acute kidney injury. Secondary outcomes were the rate of blood transfusions, time to operative intervention from admission, and length of stay. There was no standard protocol for postoperative packed red blood cell administration and the decision for transfusion was based on the attending surgeon’s professional judgment at each institution.

### 2.3. Statistical Analysis

Fishers Exact Test and t-test were used to compare the baseline characteristics and outcomes between the two study groups, with an alpha level of 0.05.

## 3. Results

### 3.1. Baseline Analysis

After reviewing 728 consecutive hip fracture cases, there were 216 hip fracture patients (29%) on chronic warfarin therapy during the study period. Sixty-one patients received both vitamin K and FFP preoperatively, while eight patients received FFP only. There were 37 total hip arthroplasty (THA) procedures performed in the low INR group, and 12 in the high INR group. The patients in the low INR group (81.29 years) were statistically significantly younger than the patients in the high INR group (83.47 years.) (*p* = 0.04).

### 3.2. Outcome Analysis

Two patients in the 1.5–3.0 group (High INR group) had a major complication event of hematoma formation (1.6%). One patient, who underwent a hemiarthroplasty procedure was treated non-operatively with medical management and observation. The other patient underwent a cephalomedullary nail and did undergo operative hematoma evacuation, thus one reoperation for hematoma (0.8%). We identified fifteen re-operations after surgical intervention with only two potentially related to bleeding complications: one superficial wound infection and one hematoma evacuation already mentioned above. The superficial wound infection occurred in a patient 6 weeks after hemiarthroplasty, who had an operative INR of 1.7; while the other patient underwent hematoma evacuation one week after a cephlomedullary nail, who had an operative INR of 2.3. The other 13 reoperations consisted of loss of fixation (7), malunion or nonunion (5), and painful hardware (1) (Table 1), which all occurred more than 30 days after the initial surgery. All other secondary outcome measures did not reach a statistical significance between the low and high INR groups (Table 2). There were no statistically significant differences in the rate of medical complications between the two groups either (Table 2).

## 4. Discussion

Orthopedic surgeons must weigh the benefits and risks of expedient surgical repair of therapeutically anticoagulated hip fracture patients. Numerous studies have identified the risks of surgical delay and the geriatric population were the most susceptible to the morbidity and mortality caused by the delay [2,3,4,5,6,8,24,25,26]. To our knowledge, this is the first investigation in the literature that evaluates the safety of surgical intervention on hip fracture patients on chronic warfarin with therapeutic INR values between 1.5 and 3.0.

Collinge et al. retrospectively reviewed the rates of post-operative bleeding complications in hip fracture patients who were either on aspirin, clopidogrel, or warfarin therapy [13]. They did not find any increase in rate of complications in their series, but they limited their patients’ sample to be those who had an operative INR value below 1.5. Similarly, to these investigations, we considered postoperative hematoma requiring operative decompression to be an important outcome measurement that captures significant morbidity for patients with therapeutic INR values. Nydick et al. reviewed post-operative complications in patients who were on clopidogrel therapy and who underwent non-elective orthopedic procedures [27]. They found no increased risks in their patients while on clopidogrel. In the case–control study, Feely et al. did not find any difference in perioperative complications, rate of blood transfusions, or mortality between hip fracture patients on clopidogrel and control patients [28]. In their retrospective comparison of hip fracture patients prescribed clopidogrel undergoing surgical intervention compared to controls, Wallace et al. found no difference in estimated blood loss. However, they did report a higher rate of blood transfusion (56% vs. 31%, *p* = 0.0121) [29]. Similar to our investigation, there was no standardized methodology regarding blood transfusion criteria in their review.

Our combined results from both institutions demonstrated no significant clinical difference in hematomas or medical complications between high and low INR groups. This is in contradistinction to other non-orthopedic procedures cited in the cardiothoracic literature demonstrating increased blood loss and transfusion in higher INR value patients [9]. We did observe two hematomas in the High INR groups; however only 1 resulted in an operative irrigation and debridement. Furthermore, no statistically significant differences were observed with regards to 30-day mortality, post-operative blood units administered, time to surgery, or length of surgery. In fact, the Low INR group had a slightly longer length of stay. presumably as a result of the INR needing to be either optimized for surgery or returned to a therapeutic level. Additionally, there were similar events of transfusions, anemia, and cardiac events.

The limitations of the study include the inherent bias and limitations of a retrospective design. As a retrospective review, our investigation reviewed established practice philosophies and tactics of multiple surgeons at two academic medical centers. This may have introduced a bias as we have an interest in justifying our existing practice tactics. Furthermore, some may believe that expeditious surgical treatment of hip fracture patients with elevated INR values is not the standard of care in their practice environment. Additionally, we did not consider the effectiveness of Vitamin K and FFP. Some but not all high-INR patients received the treatment. Moreover, we did not take a post-treatment INR value to confirm the normalization of INR value. Surgeries proceeded regardless of whether the Vitamin K and FFP actually took effect. Even though we looked at the average blood loss, the measurement of EBL as a metric is an imperfect way to evaluate blood loss anemia and its physiologic affects. Statistically, our results were limited by the unadjusted analysis. Our sample size was not powerful enough to perform a multivariate regression analysis. A prospective, multicenter trial with a power analysis would be more helpful in definitively answering this question. Patients in low and high INR groups could have differing medical optimizations and nuanced comorbid pathologies. Additionally, these results cannot be extended to newer anticoagulant medications with different mechanisms of action such as selective Xa inhibitors or direct thrombin inhibitors as these were not identified. It would be speculative to extend our results to patients anticoagulated on these medications instead of warfarin.

The delay in the surgical repair of patients with hip fractures increases the morbidity and mortality and thus surgery should be performed as soon as medically possible [4,5,6,7,8,11]. Warfarin anticoagulation has been identified as a factor contributing to delayed operative repair despite limited data in the orthopedic literature that suggest that operative INR values between 2.0 and 3.0 increase the risk profile of surgery [24]. Recognizing the danger in operative delay, some authors have written protocols to expeditiously reverse and manage therapeutically anticoagulated patients [30].

In our series, surgical repair of hip fracture patients with high INR values demonstrated no increased risk of complications, blood transfusions, or mortality. We found a relatively low rate of post-operative hematoma formation requiring drainage (0.8%) in this series. Since there are theoretical risks of delaying surgery, we feel justified in continuing our practice of acutely operating on proximal femur fracture patients with INR values of 1.5–3 but need to monitor these patients to ensure it is a safe practice. A larger-scale, prospective cohort study is necessary in order to investigate the impact of this clinical protocol for patients on chronic anticoagulants with hip fractures. Similarly, reversing therapeutic INR before hip fracture repair may be unnecessary and potentially delay surgical intervention. The management of the patient on chronic anticoagulation with a hip fracture is a complex medical and surgical event and to reduce the risks of medical and surgical complications, a better understanding of safe practice is necessary. With newer anticoagulation agents becoming more popular, understanding the risk of bleeding complications in the chronically anticoagulated patient is important for the hip fracture population.

## Figures and Tables

**Table 1 geriatrics-05-00043-t001:** Outcomes of surgical interventions on hip fracture patients (N = 216).

Outcomes	High INR Group (n = 136)	Low INR Group (n = 80)	*p*-Value
*Primary outcomes*			
Reoperations ^a^	4	2	0.89
All-type complications ^a^	28	23	0.12
Major complications	17	6	0.28
Postoperative anemia	31	8	0.02 *
*Secondary outcomes*			
Transfusions	35 (25%)	32 (41%)	0.61
Average estimated blood loss (EBL)	241 cc	195 cc	0.78
Days to operation room	1.21	1.86	0.006 *
Length of stay	6.46	8.26	0.008 *

^a^ The reoperations and complications are pertaining to conditions such as drainage of a postoperative hematoma, cardiac events (atrial fibrillation, myocardial infarction, congestive heart failure), infection (urinary tract infection, pneumonia), thromboembolic complication, or acute kidney injury within 30 days of initial surgery. * indicates statistical significance with an alpha-level of 0.05.

**Table 2 geriatrics-05-00043-t002:** Differences in the type of complications.

Complication	High INR Group (n = 136)	Low INR Group (n = 80)
Anemia	31(22.8%)	14 (17.9%)
DVT	3 (2.2%)	1 (1.3%)
PE	0 (0%)	1 (1.3%)
Pneumonia	2(1.4%)	8 ( 10.3%)
Renal Failure	4 (2.9%)	3 (3.8%)
Urinary Tract Infection	10 (7.3%)	8 (10.3%)
Cardiac Events	9 (6.6%)	5 (6.4%)
Other	9(6.6%)	5 (6.4%)

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
