# Peer review of "Postoperative Complications of Hip Fractures Patients on Chronic Coumadin: A Comparison Based on Operative International Normalized Ratio"

_geriatrics, 2020, doi:10.3390/geriatrics5030043_

Round 1
Reviewer 1 Report
This is an excellent paper with relevance in day to day trauma and orthogeriatric practice.
The findings may be somewhat surprising in certain quarters as they ask questions of routine practice. This is a welcome challenge to the norm.
Methodology is sound. Realistically this issue could only be looked at retrospectively before publication of these results.
Author Response
Thank you for the comments and recognition.
Reviewer 2 Report
The paper by Kain and colleagues explores the issue of postoperative hematological complications of patients experiencing hip fracture while on chronic therapy with warfarin. The authors retrospectively compared the surgical outcomes of individuals with normal vs. elevated INR values at 2 different trauma centers.
They hypothesized that expedient surgical intervention of hip fractures with an elevated therapeutic INR due to chronic warfarin treatment could be performed with a similar rate of morbidity and mortality compared to delayed surgery due to INR reversal.
The paper is mostly clear, and the conclusions follow logically the findings.
I have some questions to solve or revisions to suggest, which are listed below.
MAJOR ISSUES
1) The study population was divided according to the INR value: was that the value at admission, or the pre-operative one? In the Methods section the authors report that the value was “at the time of surgery”, but it is not completely clear. Otherwise, they may have collected both INR at admission and at the time of surgery. Please clarify this aspect.
2) That being said, which INR values were corrected, and which ones were not? The authors mention that both Vitamin K and FFP (I suggest writing “Fresh Frozen Plasma” the first time) were used, perhaps without a specific protocol of administration. How do I have to interpret the INR values below 1.5? Do I have to assume that all of them received either Vitamin K or FFP (or both)?
3) How many patients in this population were taking also antiplatelets agents? Please specify this data, if you can retrieve it.
MINOR CONCERNS
Page 2, line 75: This first sentence is not completely fluent, despite being clear in its significance. I suggest writing “[…] of patients who underwent surgical repair of hip fractures from [...]”.
Page 3, lines 106-107: The last part of this sentence is not fluent, it seems like a repeat term is used.
Page 3, line 117: In this last sentence the groups are defined “control” and “therapeutically anticoagulated patients”; I suggest the authors to use the same terminology across the manuscript, uniforming the definitions, e.g. “elevated INR” or “high INR” patients. These terms, indeed, have already been used at the beginning of the manuscript.
Page 4, line 126: Among the studies that explored the delay in waiting time for surgery and risk factors for mortality in the geriatric population, I suggest adding the following reference: Bellelli G, et al. J Am Med Dir Assoc 2012 Sep;13(7):664.e9-664.e14. doi: 10.1016/j.jamda.2012.06.007
Page 4, line 147: same comment as line 117 about the term “therapeutically anticoagulated”.
Page 5, line 163: same comment about “Control and therapeutically anticoagulated”. The authors just need to uniform the terms, either one or the other choice.
Page 6, lines 180-181: the part of sentence “[…] understanding how to manage patients on chronic anticoagulants with hip fractures needs close evaluation.” is not completely clear. I suggest rephrasing it.
References: please uniform the references, because some of them present the full name, others have the the first name with the initials only, etc.
Statistical revision: not necessary.
Language revision: not necessary.
Overall I think that this is a good research paper, even though the analysis is quite simple.
After addressing the comments, I think that the manuscript could improve and then be considered for publication in Geriatrics MDPI.
Author Response
[All Responses are in bold font.]
Comments and Suggestions for Authors
The paper by Kain and colleagues explores the issue of postoperative hematological complications of patients experiencing hip fracture while on chronic therapy with warfarin. The authors retrospectively compared the surgical outcomes of individuals with normal vs. elevated INR values at 2 different trauma centers.
They hypothesized that expedient surgical intervention of hip fractures with an elevated therapeutic INR due to chronic warfarin treatment could be performed with a similar rate of morbidity and mortality compared to delayed surgery due to INR reversal.
The paper is mostly clear, and the conclusions follow logically the findings.
I have some questions to solve or revisions to suggest, which are listed below.
MAJOR ISSUES
1) The study population was divided according to the INR value: was that the value at admission, or the pre-operative one? In the Methods section the authors report that the value was “at the time of surgery”, but it is not completely clear. Otherwise, they may have collected both INR at admission and at the time of surgery. Please clarify this aspect.
The INR value was collected retrospectively. Therefore, for some patients, the INR value was measured preoperatively; others at admission. We have the assumption that the INR value measured at the closest time to surgery was the INR value at the time of surgery. This was clarified in Line 86-87.
2) That being said, which INR values were corrected, and which ones were not? The authors mention that both Vitamin K and FFP (I suggest writing “Fresh Frozen Plasma” the first time) were used, perhaps without a specific protocol of administration. How do I have to interpret the INR values below 1.5? Do I have to assume that all of them received either Vitamin K or FFP (or both)?
We reported in Section 3.1 that 61 patients received both vitamin K and FFP and 8 patients received FFP only. We didn’t have a specific protocol for low INR patients. Also, we did the surgery regardless of the treatment’s taking effective. The goal was to evaluate what happened postoperatively if we operated on the patient with an INR greater than 1.5, regardless of FFP or vit K being given. Since there was not consistency with who received what the only thing we wanted to evaluate was the perioperative complications after operating on a patient with an elevated INR. We discussed this point further in Line 161-164.
3) How many patients in this population were taking also antiplatelets agents? Please specify this data, if you can retrieve it.
Unfortunately, we don’t have that information available.
MINOR CONCERNS
Page 2, line 75: This first sentence is not completely fluent, despite being clear in its significance. I suggest writing “[…] of patients who underwent surgical repair of hip fractures from [...]”.
Page 3, lines 106-107: The last part of this sentence is not fluent, it seems like a repeat term is used.
Page 3, line 117: In this last sentence the groups are defined “control” and “therapeutically anticoagulated patients”; I suggest the authors to use the same terminology across the manuscript, uniforming the definitions, e.g. “elevated INR” or “high INR” patients. These terms, indeed, have already been used at the beginning of the manuscript.
Page 4, line 126: Among the studies that explored the delay in waiting time for surgery and risk factors for mortality in the geriatric population, I suggest adding the following reference: Bellelli G, et al. J Am Med Dir Assoc 2012 Sep;13(7):664.e9-664.e14. doi: 10.1016/j.jamda.2012.06.007
Page 4, line 147: same comment as line 117 about the term “therapeutically anticoagulated”.
Page 5, line 163: same comment about “Control and therapeutically anticoagulated”. The authors just need to uniform the terms, either one or the other choice.
Page 6, lines 180-181: the part of sentence “[…] understanding how to manage patients on chronic anticoagulants with hip fractures needs close evaluation.” is not completely clear. I suggest rephrasing it.
References: please uniform the references, because some of them present the full name, others have the the first name with the initials only, etc.
All minor suggestions were addressed in the texts with suggested edits.
Statistical revision: not necessary.
Language revision: not necessary.
Overall I think that this is a good research paper, even though the analysis is quite simple.
After addressing the comments, I think that the manuscript could improve and then be considered for publication in Geriatrics MDPI.
Thank you for the comments and suggestions.
Reviewer 3 Report
This is a retrospective study comparing outcomes of patients 65 years or older with a surgically treated hip fracture with therapeutic and sub-therapeutic INR values. Eligible patients were on chronic warfarin therapy and treated at two level II trauma centers between 2005 and 2013. This is an important topic given the prevalence of older adults on anti-coagulation therapy and the increased risk of morbidity and mortality due to delayed hip fracture repair. The paper hypothesizes that expedient surgical intervention for hip fracture patients on chronic warfarin with high INR is safe. However, as written, it is unclear how the results address this hypothesis. The results are reported comparing high versus low INR with time to surgical fixation as a secondary outcome. However, if the authors want to determine if there is a difference in complication rate among patients with high versus low INR who were treated immediately, it seems like the results should be stratified by expedient and non-expedient surgery.
Specific Comments:
Abstract:
Line 37 mentions that there is no statistically significant difference in the odds of re-operation, however no logistic regression was performed.
Keywords:
Line 40: Wouldn't INR, anti-coagulation, hip fracture, and post operative complications be more appropriate? I don't see how discharge status and SNF are related.
Introduction:
- Please define sub therapeutic INR values and what is meant by “expedient” surgery.
- Line 52: The authors report the US prevalence of chronic anti-coagulation therapy but do we know the proportion of older adults with hip fracture on anticoagulants? This may be more relevant if those data exist.
- Is everyone with sub-therapeutic INR on anti-coagulation therapy? The connection between anti-coagulation therapy and INR needs to be clearer in the preceding paragraphs.
Methods
- Baseline characteristics include age, fixation type, days to surgery, INR at the time of surgery, and the use of vitamin K, but what about other potential confounders such as comorbidities and medication use?
- Secondary outcomes include rate of blood transfusions, time to operative repair and length of stay. EBL is also mentioned in table 1 but not discussed in the description of outcomes. Also, what about mortality? Seems that should be included as an outcome.
- The statistical analysis needs to be further developed. For example, "Fischer exact test and t test were used to compare age, type of fixation, days to surgery, and the use of vitamin K among patients with therapeutic INR <1.5 and INR 1.5 - 3.0." If Fischer exact test and t tests were used to compare outcomes, it is important to note that these are un-adjusted, which is a limitation of the study. Understandably, the numbers are small which challenge multivariable regression. However, it should be acknowledged.
Results
- Line 99: Provide denominator- how many patients were surgically treated for hip fracture between 2005 and 2013?
- Line 100: what is FFP?
- Recommend including a baseline table stratified by low and high INR groups; as of now there are no baseline comparisons.
- Use consistent terminology to describe the groups. Either Low and High INR or non therapeutic and therapeutic anticoagulated patients.
- Table 1- Need to define all abbreviations in the table; and make sure that each outcome is defined with sufficient detail in the methods section. For example, your primary outcome is 30 day readmission for hematoma or cardiac complication. However it's not clear from this table where that is reflected.
- The information in table 2 could be combined with the information in table 1. Not sure a second table makes much sense.
Discussion
- The first paragraph states that “this is the first investigation in the literature that reviews the safety of expedient surgical intervention on hip fracture patients with chronic warfarin with therapeutic INR values between 1.5 and 3.” If this is what you intended to study, you need to stratify results by time of surgery rather than treating time to surgery as an outcome. Was it always the site’s practice to operate immediately or did that change over time? Also, it would be worth discussing how other factors such as medication use/comorbidities and surgical fixation type are believed to impact outcomes.
Author Response
Comments and Suggestions for Authors
This is a retrospective study comparing outcomes of patients 65 years or older with a surgically treated hip fracture with therapeutic and sub-therapeutic INR values. Eligible patients were on chronic warfarin therapy and treated at two level II trauma centers between 2005 and 2013. This is an important topic given the prevalence of older adults on anti-coagulation therapy and the increased risk of morbidity and mortality due to delayed hip fracture repair. The paper hypothesizes that expedient surgical intervention for hip fracture patients on chronic warfarin with high INR is safe. However, as written, it is unclear how the results address this hypothesis. The results are reported comparing high versus low INR with time to surgical fixation as a secondary outcome. However, if the authors want to determine if there is a difference in complication rate among patients with high versus low INR who were treated immediately, it seems like the results should be stratified by expedient and non-expedient surgery.
Readdress our hypothesis
Get rid of the expedient surgery things
Specific Comments:
Abstract:
Line 37 mentions that there is no statistically significant difference in the odds of re-operation, however no logistic regression was performed.
We drew this conclusion based on crude analysis instead of a logistic regression analysis. It is because we don’t have a big enough sample size to perform a powerful regression analysis. We recognize this is an imperfect study and are just reporting from the data we have collected.
Keywords:
Line 40: Wouldn't INR, anti-coagulation, hip fracture, and post operative complications be more appropriate? I don't see how discharge status and SNF are related.
Thank you for pointing it out, and agree. We made a mistake in the keywords list. Now, we corrected the list to reflect the relevant information.
Introduction:
- Please define sub therapeutic INR values and what is meant by “expedient” surgery.
A definition for subtherapeutic INR is provided. “Expedient” is in reference to delayed surgery; however, we deleted this term in order to avoid confusion, as we are really evaluating how patients do if they are operated on regardless of their INR value.
- Line 52: The authors report the US prevalence of chronic anti-coagulation therapy but do we know the proportion of older adults with hip fracture on anticoagulants? This may be more relevant if those data exist.
It has been estimated that 4-8% of patients are on chronic coumadin. (ref: Al-Rashid M, Parker MJ. Anticoagulation management in hip fracture patients on warfarin. Injury 2005;11:1311-1315. [PubMed] [Google Scholar]). Also in our study, we identified 216 patients our 728 pts screened which means we had 30% of our hip fracture patients on coumadin (Line 61-62 and in results, Line 110)
- Is everyone with sub-therapeutic INR on anti-coagulation therapy? The connection between anti-coagulation therapy and INR needs to be clearer in the preceding paragraphs.
All patients in our study cohort were under chronic anticoagulation therapy. The group with low INR were subtherapeutic.
Methods
- Baseline characteristics include age, fixation type, days to surgery, INR at the time of surgery, and the use of vitamin K, but what about other potential confounders such as comorbidities and medication use?f
Since all patients were on chronic coumadin for pre-existing conditions, all patients were either an ASA score of 3 or 4. This is a limitation, as it is not well defined but with assumption that the all have some condition requiring coumadin we were just looking at operative INR value.
- Secondary outcomes include rate of blood transfusions, time to operative repair and length of stay. EBL is also mentioned in table 1 but not discussed in the description of outcomes. Also, what about mortality? Seems that should be included as an outcome.
We fell That the ebl is just an estimate and that since this was not consistenly measured it was not a meaningful data point.
- The statistical analysis needs to be further developed. For example, "Fischer exact test and t test were used to compare age, type of fixation, days to surgery, and the use of vitamin K among patients with therapeutic INR <1.5 and INR 1.5 - 3.0." If Fischer exact test and t tests were used to compare outcomes, it is important to note that these are un-adjusted, which is a limitation of the study. Understandably, the numbers are small which challenge multivariable regression. However, it should be acknowledged.
It is true that we don’t have enough number of patients to perform an adjusted regression analysis. This is added to the limitations. (Line 167-169)
Results
- Line 99: Provide denominator- how many patients were surgically treated for hip fracture between 2005 and 2013? 728
- Line 100: what is FFP?
FFP is stand for fresh frozen plasma. The full spelling was added where it is first mentioned in Line 86.
- Recommend including a baseline table stratified by low and high INR groups; as of now there are no baseline comparisons.
- Use consistent terminology to describe the groups. Either Low and High INR or non therapeutic and therapeutic anticoagulated patients.
We consolidated the use of terminology.
- Table 1- Need to define all abbreviations in the table; and make sure that each outcome is defined with sufficient detail in the methods section. For example, your primary outcome is 30 day readmission for hematoma or cardiac complication. However it's not clear from this table where that is reflected.
The abbreviations were spelled out, and footnotes were added.
- The information in table 2 could be combined with the information in table 1. Not sure a second table makes much sense.
We think it is clearer to separate two tables.
Discussion
- The first paragraph states that “this is the first investigation in the literature that reviews the safety of expedient surgical intervention on hip fracture patients with chronic warfarin with therapeutic INR values between 1.5 and 3.” If this is what you intended to study, you need to stratify results by time of surgery rather than treating time to surgery as an outcome. Was it always the site’s practice to operate immediately or did that change over time? Also, it would be worth discussing how other factors such as medication use/comorbidities and surgical fixation type are believed to impact outcomes.
We appreciate the questions and agree this needs to be clarified. This practice evolved over time and we realized we had several approaches to the problem going on and wanted to evaluate the perioperative complications of ignoring the INR. We collaborated with another institution to increase our numbers since they had similar experience. This is not about the expedient nature of the surgery, but rather about trying to assess if ignoring the INR value, if below 3, is safe. By ignoring the INR value and not full reversing we thought we would be avoiding unnecessary delays. This is not a definitive study rather some information to add to the literature that exists and report the study we did.
Line 145-147: We have changed the sentence to: “To our knowledge, this is the first investigation in the literature that evaluates the safety of surgical intervention on hip fracture patients on chronic warfarin with therapeutic INR values between 1.5 and 3.0.”
Round 2
Reviewer 3 Report
I appreciate that the authors addressed many of my comments and suggestions which I believe has improved the quality of the paper. However, there are some additional edits that I think are important to address:
Abstract:
Line 37 mentions that there is no statistically significant difference in the odds of re-operation, however no logistic regression was performed. I would remove the term “odds of re-operation” and replace with “rate of complication”.
Introduction
I understand now that the goal was to compare outcomes of surgically treated hip fracture among patients with high versus low INR, regardless of timing of surgery. Please consider re-framing your hypothesis to something like: “We hypothesized that there would be no difference in complication rate following surgical intervention for patients with therapeutic versus sub-therapeutic INR values."
Methods
- I would include length of stay and time to operation as baseline characteristics rather than outcomes.
- If EBL is not an important outcome, either remove it or include it as a baseline characteristic.
- Mortality is not addressed in the methods or the results, though it is mentioned in the hypothesis statement.
- I recommend updating the statistical analysis section to read "Fischer exact test and t test were used to compare baseline characteristics and outcomes among patients with therapeutic INR <1.5 and INR 1.5 - 3.0."
Results
- Recommend including a baseline table stratified by low and high INR groups comparing age, gender, type of fixation, time to surgery, LOS, FFP, and K.
- Compare rates of mortality (or remove this from the hypothesis statement)
Discussion
- Line 162- remove reference to control group; refer to high/low INF.
Author Response
Comments and Suggestions for Authors
I appreciate that the authors addressed many of my comments and suggestions which I believe has improved the quality of the paper. However, there are some additional edits that I think are important to address:
Abstract:
Line 37 mentions that there is no statistically significant difference in the odds of re-operation, however no logistic regression was performed. I would remove the term “odds of re-operation” and replace with “rate of complication”.
Odds is not unique to logistic regression. Even though we didn’t perform an adjusted regression analysis, we did a crude comparison on the odds of reoperation between the two study groups. Therefore, odds and rate is interchangeable here.
Introduction
I understand now that the goal was to compare outcomes of surgically treated hip fracture among patients with high versus low INR, regardless of timing of surgery. Please consider re-framing your hypothesis to something like: “We hypothesized that there would be no difference in complication rate following surgical intervention for patients with therapeutic versus sub-therapeutic INR values."
The hypothesis sentence was changed accordingly. (Line 71-73)
Methods
I would include length of stay and time to operation as baseline characteristics rather than outcomes.
If EBL is not an important outcome, either remove it or include it as a baseline characteristic.
Mortality is not addressed in the methods or the results, though it is mentioned in the hypothesis statement.
I recommend updating the statistical analysis section to read "Fischer exact test and t test were used to compare baseline characteristics and outcomes among patients with therapeutic INR <1.5 and INR 1.5 - 3.0."
We consider length of stay, time to operation, and EBL as secondary outcomes, because they occurred during or after the surgical intervention. They are not baseline characteristics per se.
We changed the sentence describing statistical analysis in Line 99-100.
Results
Recommend including a baseline table stratified by low and high INR groups comparing age, gender, type of fixation, time to surgery, LOS, FFP, and K.
Compare rates of mortality (or remove this from the hypothesis statement)
We decided to not include a baseline table. We removed mortality from the hypothesis.
Discussion
Line 162- remove reference to control group; refer to high/low INF.
There is no reference to “control group” in Line 162. However, we re-worded the sentence in Line 151 to get rid of the control and therapeutic groups.